# Facile orientation control of MOF-303 hollow fiber membranes by a dual-source seeding method

Mengjiao Zhai[1,2], Farhad Moghadam[1,2], Tsaone Gosiamemang[2], Jerry Y. Y. Heng ●[2] & Kang Li ●[1,2] ✉

Metal–organic frameworks (MOFs) are nanoporous crystalline materials with enormous potential for further development into a new class of high-performance membranes. However, the preparation of defect-free and water-stable MOF membranes with high permselectivity and good structural integrity remains a challenge. Herein, we demonstrate a dual-source seeding (DS) approach to produce high-performance, water-stable MOF-303 membranes with hollow fiber (HF) geometry and preferentially tailored crystallographic orientation. By controlling the nucleation site density during secondary growth, MOF-303 membranes with a preferred crystallographic orientation (CPO) on the (011) plane were fabricated. The MOF-303 membrane with CPO on (011) provides straight one-dimensional permeation channels with a superior water flux of 18 kg m$^{-2}$ h$^{-1}$ in pervaporative water/ethanol separation, which is higher than that of most of the reported zeolite membranes and 1–2 orders of magnitude greater than that of previously reported MOF membranes. The straight water permeation channels also offer a promising water permeance of 15 L m$^{-2}$ h$^{-1}$ bar$^{-1}$ and a molecular weight cut-off (MWCO ≈ 269) for dye nanofiltration. These results provide a concept for developing ultra-permeable MOF membranes with good selectivity and structural integrity for pervaporation and nanofiltration.

As a direct consequence of climate change and population growth, water scarcity is currently one of the most serious crises. Limited access to potable water has threatened the health of more than a billion people globally[1]. The growing need for healthy water necessitates technological innovations for the development of sustainable and energy-efficient water purification systems[1]. Compared with conventional water separation processes such as evaporation, adsorption or crystallization, membrane technology is an energy-efficient and environmentally benign process widely used in wastewater treatment, desalination, etc.[2–5] The applicability of commercially available polymeric membranes is generally constrained by the intrinsic trade-off between water permeability and water-solute selectivity. Therefore,

the fabrication of high-performance polymeric membranes with excellent permselectivity is still challenging[6]. Thus, the design and synthesis of new nanoporous materials and their membranes to overcome this trade-off are urgently needed.

MOFs are nanoporous crystalline materials with tunable pore sizes and versatile chemical functionalities, demonstrating great potential for membrane-based separation[6–10]. Despite progress in gas separation applications, their potential in aqueous media remains limited because water molecules can induce hydrolysis by attacking metal–organic bonds and replacing ligands, altering the crystal phase and causing structural failure[11–13]. According to the hard–soft–acid–base (HSAB) theory[14], stable MOFs are expected to be generated by

[1]Barrer Centre, Chemical Engineering Department, Imperial College London, London, UK. [2]Department of Chemical Engineering, Imperial College London, London, UK. ✉e-mail: kang.li@imperial.ac.uk

coordinating high-valence metal ions ($Fe^{3+}$, $Al^{3+}$, $Zr^{4+}$, etc.) with carboxylate-type ligands or soft azolate linkers (imidazolate, pyrazolates, etc.) with divalent metal ions ($Zn^{2+}$, $Cu^{2+}$, $Ni^{2+}$, $Co^{2+}$), resulting in strong metal–linker coordination[11,13]. By surpassing the affinity of metal ions for water molecules, strong metal–linker coordination bonds are highly resistant to hydrolysis. Following this strategy, Zr-based UiO-66 was the first MOF to demonstrate good water stability and separation performance in desalination[15] and dye nanofiltration[16]. Another example is a recently reported Al-based MOF, i.e., MOF-303 [Al(OH)(PZDC), PZDC = 1H-pyrazole-3,5-dicarboxylate], in which the strong metal–linker coordination between high-valent $Al^{3+}$ and carboxylate groups confers high structural stability. Additionally, the pyrazole groups in MOF-303 serve as functional sites, providing high water sorption capacity and facilitating rapid water sorption–desorption cycles, making it a promising candidate for water harvesting applications[17–20]. MOF-303 comprises one-dimensional (1D) channels with a 0.6 nm pore size and hydrophilic characteristics due to the PZDC linker, making it a suitable nanoporous material for developing high-performance water purification membranes[21,22]. Cong et al.[21] fabricated a MOF-303 membrane via in situ growth of MOF-303 crystals on a flat alumina disc, which showed good divalent ion rejection but very low water permeance (<0.8 L m$^{-2}$ h$^{-1}$ bar$^{-1}$) because the membrane was prepared with a high thickness and random orientation of MOF crystals. Lai et al.[22] used a seeding growth technique to prepare MOF-303 membranes with very high water/ethanol selectivity, but the water flux ranged from 0.1–0.2 kg m$^{-2}$ h$^{-1}$ because of the partially amorphous structure of the crystals. While these studies demonstrate good water stability, the membranes still suffer from low water permeance owing to their high thickness for patching intercrystalline defects.

A major challenge contributing to low permeance is the random orientation of MOF crystals, which considerably undermines the intrinsic features of MOFs in three-dimensional (3D) structures. Controlling the MOF crystal orientation is an effective way to fabricate MOF membranes with high permeance and good size sieving performance[23]. Additionally, when making MOF-based composite membranes, the formation of interfacial defects between the MOF membrane and the substrate is another issue that is detrimental to selectivity. This is because, in general, neither the metal component nor the organic linker provides a link to bond to the support surface, making the membranes structurally unstable. The reactive seeding (RS) technique reported by Hu et al.[24,25] is a facile secondary growth method in which an alumina support can be used as a metal source to interact with a linker to form high-density MOF seeds, which not only mitigates interfacial defects but also considerably minimizes the formation of intercrystalline voids, thus improving MOF membrane selectivity. Therefore, to make high-performance MOF membranes, a fabrication technique that can (i) minimize intercrystalline defects, (ii) improve the interfacial compatibility between the MOF layer and substrate and (iii) control the orientation of MOF permeation channels should be developed.

In this study, we propose a DS method to achieve the above goal in the preparation of high-performance MOF-303 membranes on an alumina HF substrate. The alumina substrate acts as a metal source to react with organic linkers to form seeds, which results in better interfacial interactions between MOF crystals and the HF substrate. The second metal source provided by the synthesis solution allows the density of seeds to be preferentially adjusted. The subsequent secondary growth of crystals with a tailored orientation eliminates intercrystalline defects induced by random nucleation in the seeding step. By tailoring the orientation of the MOF-303 crystals, an unprecedented water flux, i.e., up to two orders of magnitude greater than that of previously reported MOF-based membranes with promising selectivity, was achieved for the membrane with CPO on the (011) for ethanol dehydration and dye nanofiltration applications. The crystal growth mechanism and orientation of the MOF-303 membranes were also

thoroughly investigated via X-ray diffraction (XRD) and scanning electron microscopy (SEM) analyses. The proposed study provides a concept for developing ultrahigh water-permeable MOF membranes with good selectivity and structural integrity for pervaporation and nanofiltration.

## Results

### Fabrication and characterization of MOF-303 HF membranes

A DS approach illustrated schematically in Fig. 1 was used to fabricate the MOF-303 HF membrane. In this method, metal sources from both the alumina HF substrate and the seeding solution, i.e., aluminum chloride hexahydrate ($AlCl_3 \cdot 6H_2O$), contribute to the formation of seeds. The alumina octahedral structure on the surface of the alumina HF substrate was first formed in an aqueous solution containing NaOH at high temperature[26] and then coordinated with ligands (1H-pyrazole-3,5-dicarboxylic acid [$H_3PDC$]) in the reaction solution to generate MOF-303 seeds on the surface of the support[24,27], illustrated as the red dots in Fig. 1. In the secondary growth step, the MOF-303 seeds further grew to the final MOF-303 membrane, which was named the DS membrane. For comparison, MOF-303 membranes were also produced using conventional RS and one-step hydrothermal methods (Supplementary Figs. 1, 2)

The morphology and crystalline structure of the MOF-303 membranes prepared by a DS approach were investigated using SEM and XRD, as shown in Fig. 2. After the seeding step, the seeds formed a thin layer almost fully covering the surface of the support, as shown in Fig. 2a, b. In the DS method, seeds grow from the grain boundaries of the support, encapsulating the alumina particles and ultimately forming the seeding layer. This distinctive growth pattern is illustrated in Fig. 2a, b, and the inset in Fig. 2b. The metal sources provided by both the seeding solution and the alumina support account for this growth pattern. The deficiency in the stoichiometric ratio of metal in the solution allows for the participation of the metal source from the alumina support, facilitating the nucleation of the alumina grains. Promoted by the ligand and metal source in the synthesis solution, the nuclei consistently grow from the alumina grains towards the nutrient-rich side (i.e., the outer surface of the HF), leading to the formation of seeds and establishing a strong interaction between the seeding layer and the support. Simultaneously, the reaction between the metal source and ligand in the synthesis solution also contributes to seed formation, thereby increasing the density of the seeding layer.

In contrast, in the conventional one-step process, the metal consumption in the solution is expected to be primary since it is sufficient and readily available for reactions, resulting in the instant formation of crystals in the solution and on the surface of the support (Supplementary Fig. 4b–e). This leads to a weak attachment between the seeding layer and the support.

After 1 h of secondary growth, the seeding layer formed a MOF-303 membrane. However, intercrystalline defects were observed on the surface of the membrane due to insufficient growth time (highlighted with red circles in Fig. 2c, d). These intercrystalline defects disappeared after 48 h of secondary growth when a uniform membrane with a thickness of 2.6 μm formed (Fig. 2e, f), signifying successful intergrowth of the crystals. Notably, the crystals produced using the DS approach exhibited a preferred orientation, growing vertically from the substrate, particularly on the upper part of the membrane (Fig. 2f and Supplementary Fig. 3a). This observation aligns well with the XRD results presented in Fig. 2g. The random crystal orientation of the seeding layer was confirmed by the (011), (022), and (1-2-2) peaks, with the (011) peak showing stronger intensity. During the 1-h secondary growth, the (011) and (022) peaks became increasingly stronger and sharper due to the growth of the crystals. In particular, a peak corresponding to the (111) face appeared, suggesting that random crystal growth still occurred at this stage, resulting in the random orientation of the crystals (Fig. 2d and Supplementary Fig. 3b).

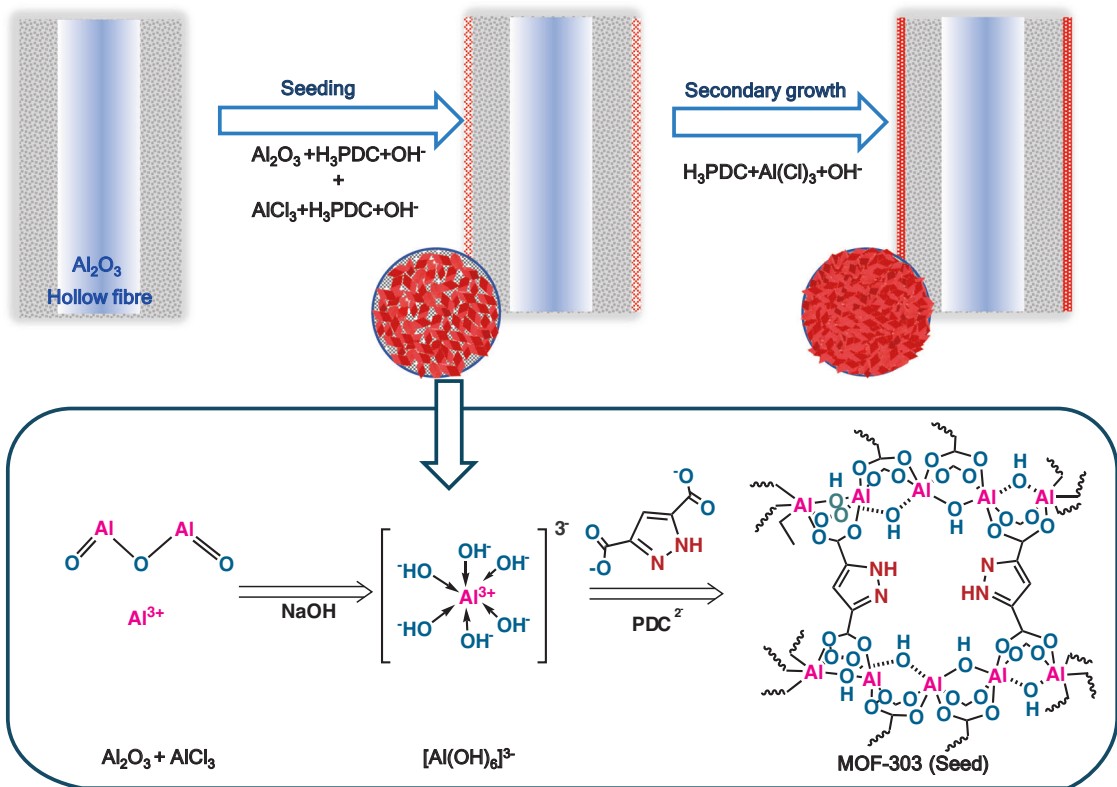

**Fig. 1 | Schematic illustration of MOF-303 membranes synthesized by a DS method.** The alumina HF is placed vertically in an autoclave filled with a ligand solution containing NaOH and an inadequate stoichiometric amount of aluminium chloride hexahydrate (AlCl₃·6H₂O). The ligand reacts with the OH groups both on the surface of the alumina HF support and in the synthesis solution to form the seed layer (seeding step), followed by a secondary growth for the final MOF-303 membranes named as DS membrane.

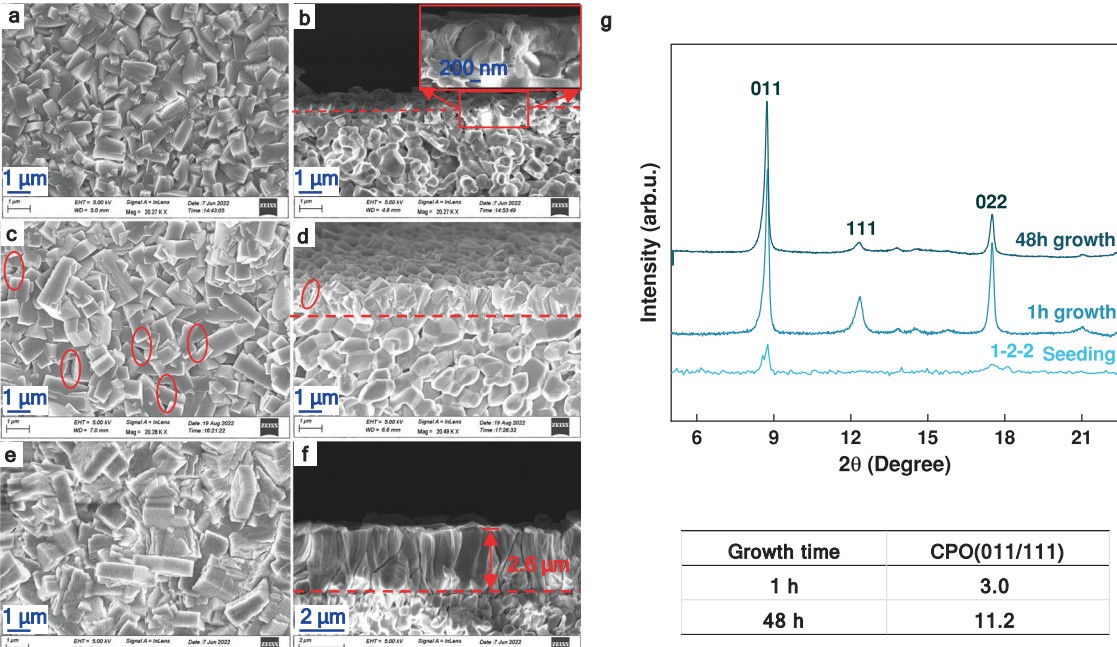

**Fig. 2 | SEM images and XRD patterns of MOF-303 membranes fabricated by DS method.** SEM images of the membrane after (**a**, **b**), seeding (the inset in (**b**) shows an enlarged view of the seeding layer covering the alumina HF substrate), (**c**, **d**) 1-h secondary growth, and (**e**, **f**) 48-h. **a**, **c**, **e** are surface images, and (**b**, **d**, **f**) are cross-sectional images of the membrane. The red circles in (**c**) display the intercrystalline gaps. The red dotted lines in (**b**, **d**, **f**) divide the cross-sections into the HF substrate and MOF layer parts. **g** XRD patterns and CPO ratios of the membrane.

With the extension of the reaction time to 48 h, the relative intensity of the (111) peak gradually decreased, and that of $CPO_{011/111}$ increased from 3 to 11, indicating preferable growth along with (011).

The crystal orientation in the DS membranes can be explained based on the evolutionary selection model proposed by van der Drift[3]. According to this model, all the seed crystals are expected to grow randomly at the same face-dependent rate in the initial stage[28]. This stage continues until the seeds meet neighboring crystals. Then, crystals with faster growth in the out-of-plane or nearly out-of-plane direction ultimately overgrow other crystals, dominating the orientation of the top membrane layer. This theory determines the crystal growth orientation if (i) the crystals follow an anisotropic growth pattern, (ii) all the crystals in the seed layer grow simultaneously, and (iii) the seed layer contains competing crystals in close contact. Given the anisotropic structure and high density of MOF-303 crystals in the seeding layer, the evolutionary selection model perfectly explains the out-of-plane orientation (vertical alignment) of the DS membrane. This is supported by the initial random orientation and the preferable orientation on the top part of the membrane (Fig. 2d–f, and Supplementary Fig. 3).

To further demonstrate the DS concept, the conventional RS approach was employed to fabricate the MOF-303 membrane for comparison. As shown in Fig. 3a, b, MOF-303 particles were seeded at the grain boundaries and grew to fill the gaps between the grains. This phenomenon aligns with the theory that crystal growth is favored on rough surfaces compared with smooth surfaces[29]. Since no metal salt was added in the seeding step, it can be assumed that these particles (clusters) were formed due to the reaction between the metal source from the alumina substrate and the $H_3PDC$ ligand. The formation of multiedge clusters in Fig. 3a, b can be attributed to either the accumulation of small crystal seeds in the rough corners of the surface or the formation of crystals due to the imperfect stoichiometric ratio. Both cases are shown to induce twinning during the growth step[30,31]. No continuous membrane was formed after 1 h of secondary growth because the intergrowth of the crystals was not complete (Fig. 3c, d). According to classic nucleation theory[32], subsequent growth from

these seeds is favored over the formation of new crystals because of the lower nucleation barrier. After 48 h of secondary growth, a uniform RS membrane with well-intergrown crystals fully covering the substrate surface and a thickness of approximately 3.5 μm was formed, as shown in Fig. 3e, f. The RS membrane has a grass-like morphology, originating from a single root (seed) on the alumina surface and extending in all directions (Fig. 3d). These crystal clusters originate from stacked tiny seeds and extend from the support surface until they meet their neighboring crystals, forming a continuous membrane containing twin crystals, as shown in Fig. 3f. Although the precise effect of crystal twinning on membrane properties remains complex and depends on various factors, it can lead to pore obstruction[33]. The XRD patterns of the samples after 1-h and 48-h secondary growth are shown in Fig. 3g, where the (011) and (1-2-2) planes become dominant after 1 h of secondary growth, with a stronger intensity for the latter. By increasing the secondary growth time to 48 h, the relative intensity of the (1-2-2) and (011) reflections, $CPO_{1-2-2/011}$, decreased to less than half of its initial value, implying that the crystals tended to grow in the (011) direction but that (1-2-2) was still the dominant growth direction of the RS membrane.

Considering the very low density of stacked metastable crystals in the RS seeding stage (Fig. 3a), the Bons and Bons model[34] describes the "grass-like" morphology well. Rooted in scattered small seeds, a divergent crystal structure emerged in all free directions during the secondary growth step (Fig. 3d) until it eventually met and connected with neighboring crystals to form the RS membrane, as shown in Fig. 3f. The lateral growth is ultimately confined to neighboring crystals and gradually directed toward the out-of-plane orientation, as verified by the decrease in $CPO_{1-2-2/011}$ from 15.7 to 7.

To further demonstrate the advantages of our DS approach, the MOF-303 HF membrane was also fabricated using the conventional one-step method, and its SEM and XRD results are presented in Supplementary Fig. 4. As shown in Supplementary Fig. 4a, the most significant peak reflected by the (011) peak matches that of the MOF-303 powder (Supplementary Fig. 5). The CPO indices of the (1-2-2) to (011) reflections ($CPO_{1-2-2/011}$) remained constant, implying that the crystal

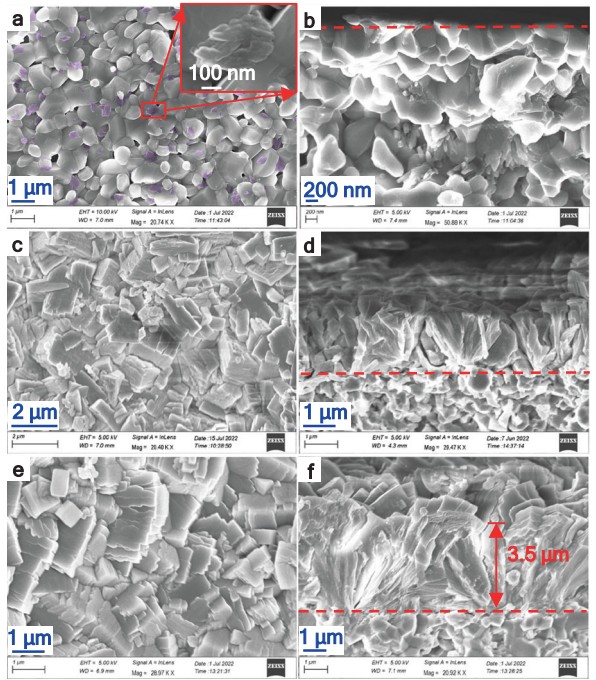
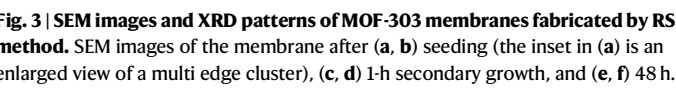
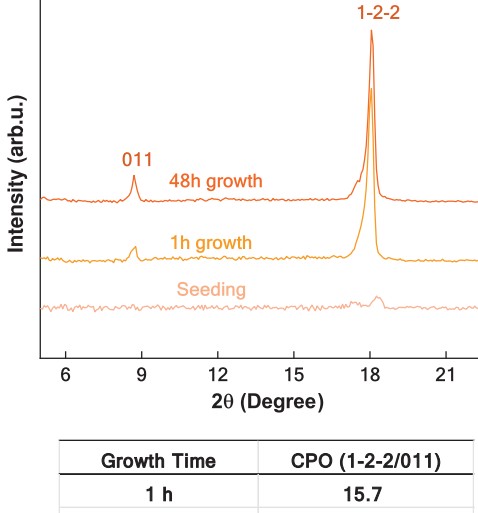

| Growth Time | CPO (1-2-2/011) |
|---|---|
| 1 h | 15.7 |
| 48 h | 7.0 |

**Fig. 3 | SEM images and XRD patterns of MOF-303 membranes fabricated by RS method.** SEM images of the membrane after (**a**, **b**) seeding (the inset in (**a**) is an enlarged view of a multi edge cluster), (**c**, **d**) 1-h secondary growth, and (**e**, **f**) 48 h. **a**, **c**, **e** are surfaces, and (**b**, **d**, **f**) are cross-sections of the membrane. The red dotted lines in (**b**, **d**, **f**) divide the cross-sections into alumina substrate and MOF-303 membrane parts. **g** XRD patterns and CPO ratios of the membrane.

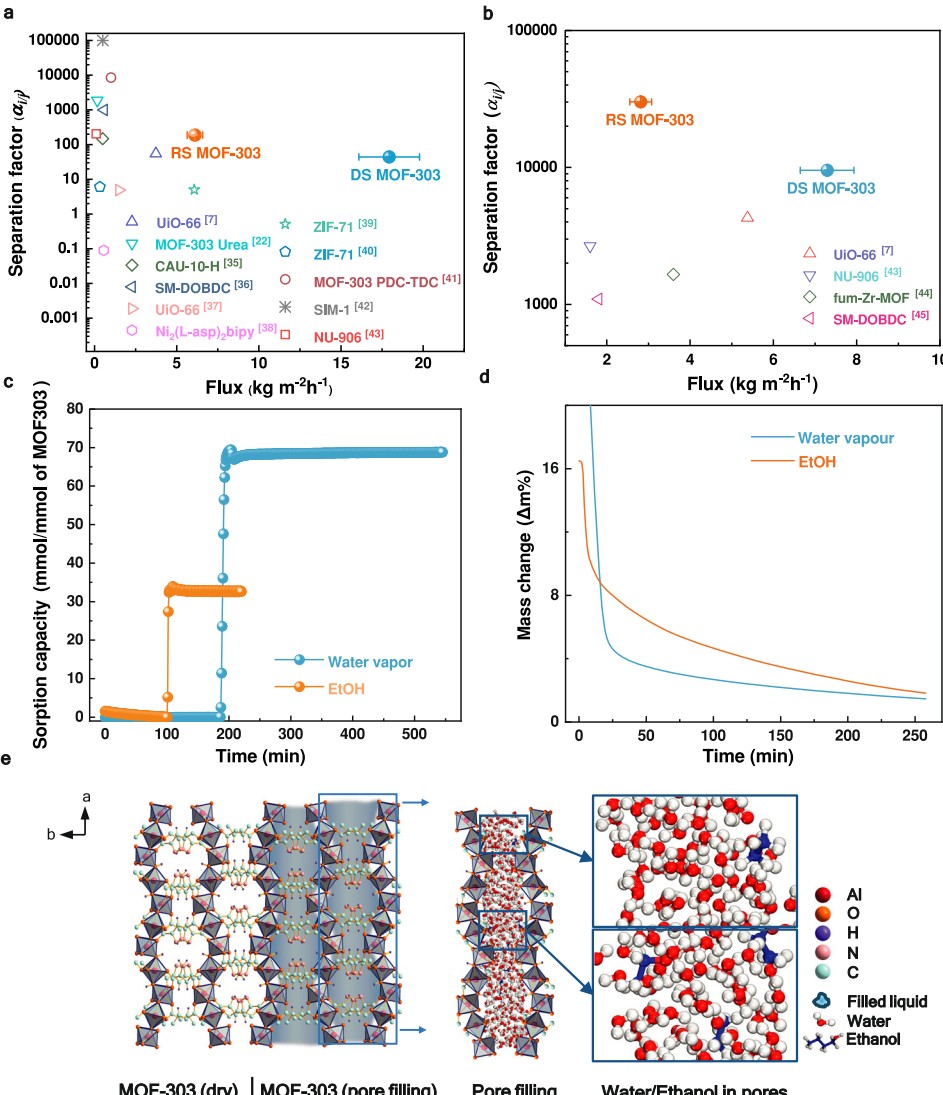

**Fig. 4 | Pervaporative water/alcohol separation of the MOF-303 membranes.**
**a** Water/ethanol (5/95, wt%) and (**b**) water/butanol (5/95, wt%) separation performances of the RS and DS membranes compared with those of other MOF-based membranes. The feed temperature was 75 °C. Data are presented as mean standard deviation values, $n$ = 3 or 4. **c** Water and ethanol sorption kinetics of MOF-303 powder measured by DVS at P/Psat = 90% and $T$ = 60 °C. **d** Desorption rates of pure water and pure ethanol measured by DVS. **e** Side view of (left) the pristine MOF-303 crystal and (right) MOF-303 in terms of dynamic pore filling in the pervaporation process. The pores contain water and ethanol molecules, but they are mainly occupied by water molecules. The higher concentration of water molecules in the pores results in a diffusion barrier to ethanol molecules because of their larger kinetic diameter. Source data are provided as a Source Data file.

orientations were identical over the growth time. Despite the formation of crystalline structures, intercrystalline defects were observed via SEM, as shown in Supplementary Fig. 4, especially in the cross section, as highlighted by the red circles (Supplementary Fig. 4e). The formation of these intercrystalline defects is likely due to inefficient heterogeneous nucleation on the support during one-step hydrothermal growth.

## Pervaporative alcohol dehydration performance

High water sorption capacity and rapid water adsorption and desorption rates are the key features of water-stable MOF-303. These results indicate the preferential permeation of water in alcohol dehydration. Both the DS and RS MOF-303 membranes demonstrated good ethanol and 1-butanol dehydration performances in a pervaporation system, as shown in Fig. 4a, b. For a feed containing 95 wt% ethanol, the fluxes of the DS and RS membranes were 18 and 6 kg m⁻² h⁻¹, respectively, which are greater than those of most of the state-of-the-art

membranes and 1–2 orders of magnitude greater than those of previously reported MOF and zeolite membranes, as shown in Supplementary Table 1 and Fig. 4a[7,22,35–43]. Compared with those of ethanol, the MOF-303 membranes prepared in this study presented a greater separation factor for butanol dehydration compared to the data given in Supplementary Table 2[22,43–45]. This can be ascribed to the larger kinetic diameter and more hydrophobic nature of butanol, which also caused the flux difference between ethanol/water and butanol/water, as illustrated in Supplementary Fig. 7. To the best of our knowledge, the water fluxes of the DS and RS membranes are among the highest values reported thus far for state-of-the-art membranes. Interestingly, the water flux of the DS membrane is approximately 200% greater than that of the RS membrane because of the straight water permeation channels provided by the vertically aligned MOF-303 crystals. On the other hand, the RS membrane offered a slightly greater water/ethanol separation factor because the divergent MOF-303 crystals, which are partially blocked, created tortuous permeation pathways.

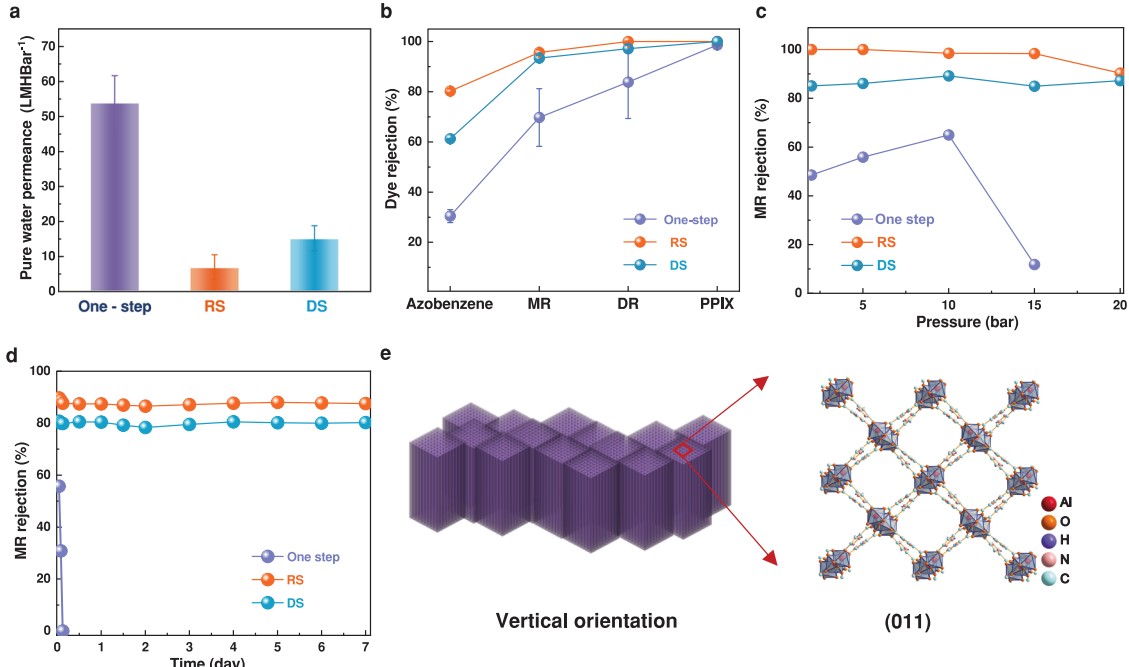

**Fig. 5 | Dye nanofiltration performance of MOF-303 membranes. a** Pure water permeance and (**b**) dye rejection performance of MOF-303 membranes fabricated via one-step growth, the RS method, and the DS method; Neutral dyes with different molecular weights were used in this study (azobenzene: 182.22 g mol⁻¹, MR: 269.3 g mol⁻¹, DR: 324.3 g mol⁻¹, PPh-IX: 562.66 g mol⁻¹). Data are presented as mean standard deviation values, $n = 3$ or 6. **c** MR rejection performance of MOF-303 membranes at different pressures. **d** Long-term stability of MOF-303 membranes at a pressure of 5 bar. **e** Schematic illustration of the DS membrane with vertical one-dimensional channels. Source data are provided as a Source Data file.

A comparison of the kinetic diameters of water vapor (2.9 Å) and ethanol (4.3 Å) molecules[44] with the pore size of MOF-303 crystals (6.0–8.0 Å), as shown by the Brunauer-Emmett-Teller (BET) test results in Supplementary Fig. 6, implies that diffusion selectivity on the basis of pore size is not the rate-determining step in water/ethanol separation. According to a previous study, both the adsorption capacity and diffusion rate determine the water and ethanol permeation rates through MOF-303 membranes. An extraordinary water affinity was visually confirmed by observing the water contact behavior shown in Supplementary Movie 1, wherein the droplet traversed the membrane in a mere 1-second interval. This high water affinity was further demonstrated by measuring the water vapor sorption kinetics of MOF-303 powder using a dynamic vapor sorption (DVS) system (Methods and Fig. 4c). The adsorption of ethanol also occurs in the pervaporation process. However, with only one hydrophilic end, ethanol exhibited a lower affinity for MOF-303, as shown in Supplementary Movie 2, where penetration took more than twice as long as water. The DVS results also indicated that the ethanol sorption capacity was less than a quarter that of water (Fig. 4c). Due to the -OH and -NH groups in the pyrazole linker, the hydrophilic nature of the MOF-303 membrane traps the first water molecule with three hydrogen bonds, which seeds for the formation of trimeric clusters. These clusters act as the nucleus for binding other water molecules and form a water network throughout the pores[17], leading to continuous pore filling of the MOF (Fig. 4e)[46] and thus ultrahigh water flux.

In dynamic pore filling, water molecules are expected to dominate the pores. These water molecules can readily diffuse through the pores by sliding over those strongly attached to the pore wall (surface diffusion). The water molecules in the pores act as a significant diffusion barrier to ethanol and 1-butanol with larger kinetic diameters, thus ensuring an ample supply of water molecules for subsequent desorption[47].

The desorption rates of water and ethanol can be affected by their diffusion through pores. Due to the higher concentration of water in the pores, its diffusion driving force is greater than that of ethanol, resulting in a faster desorption rate for water. In addition, the diffusion barrier to ethanol molecules with one hydrophobic end and a larger kinetic diameter is greater than that to water. This was confirmed by comparing the desorption rates of pure water and ethanol measured by DVS, in which ethanol exhibited a slower desorption rate despite its higher vapor pressure (Fig. 4d).

Therefore, such good water/ethanol separation performance could be a combined effect of high water adsorption, hindered diffusion of ethanol in the water-filled pores and higher desorption rates through the pores of MOF-303 membranes.

## Performance in dye nanofiltration

The dye nanofiltration performance of the MOF-303 HF membranes was evaluated to gain further insight into their size-sieving ability. The pure water permeance of the MOF-303 HF membranes was measured using a dead-end filtration system. The membrane prepared via the one-step method demonstrated pure water permeance up to 54 L m⁻² h⁻¹ bar⁻¹ but very poor rejection, largely due to intercrystalline defects[48] (Fig. 5a, b), where the molecular weight cut-off (MWCO) was tested using four neutral dye molecules, namely, azobenzene, methyl red (MR, 269.3 g mol⁻¹), disperse red (DR, 324.3 g mol⁻¹), and proto-porphyrin IX (PPh-IX, 562.7 g mol⁻¹). As shown, the MWCO of the RS and DS membranes was approximately 269.3 g.mol⁻¹, whereas the membrane fabricated by the one-step approach showed poor rejection performance (MWCO ≈ 562.7; Fig. 5b). The size-sieving mechanism, rather than adsorption, was confirmed by the increased dye concentration in the retentate, as shown in Supplementary Fig. 8. The water permeances of the RS and DS membranes were 7 and 15 L m⁻² h⁻¹ bar⁻¹, respectively, remarkably surpassing those reported for MOF-303 membranes fabricated on flat alumina discs (0.8 L m⁻² h⁻¹ bar⁻¹)[21] and those prepared by direct liquid contact and vacuum coating seeded growth methods (Experimental Section and Supplementary Figs. 9–11). This significant increase in water

permeance and improved selectivity result from the thin and uniform thickness of the membrane and the controlled orientation of the permeation channels.

As schematically illustrated in Fig. 5e, the (011) orientation of the crystals prepared via the DS method was well-intergrown such that the number of grain boundary defects could be effectively minimized. This is in good accordance with the SEM images shown in Fig. 2e, f, especially compared with the one-step membrane presented in Supplementary Fig. 4, where significant gaps and pinholes tend to form at the grain boundaries in the membranes. These defects, which are larger than intracrystalline pores, create non-MOF transport pathways that enhance flux but compromise molecular sieving. Furthermore, as an anisotropic MOF with one-dimensional pores, the (011) orientation provides shorter mass transport pathways aligned perpendicular to the substrate, facilitating straight water permeation channels with reduced transport resistance (Fig. 5e)[17]. Conversely, while the RS membrane also demonstrated high selectivity due to well-intergrown crystals, the inclined orientation and partially blocked pores from neighboring crystals created longer, more tortuous permeation channels, resulting in significantly lower water permeance.

The durability of the membranes was evaluated under different pressures and over a prolonged period. Both the RS and DS MOF-303 membranes demonstrated stable water permeance and good compaction resistance and maintained good MR dye rejection under pressures of up to 20 bar, whereas the membrane fabricated via the one-step technique could not withstand pressures greater than 10 bar (Fig. 5c and Supplementary Fig. 12). To further prove the good structural integrity of the membranes, the long-term MR rejection performance was investigated for 7 days (Fig. 5d). Unlike the one-step fabricated membrane, the RS and DS MOF-303 membranes demonstrated stable performance without losing rejection. This stable performance is attributed to (i) the better attachment of MOF-303 crystals to the support in the seeding step realized by the metal source available on the HF substrate, (ii) the higher density of MOF-303 crystals in the seeding layer, and (iii) the good growth of the crystalline structure. Furthermore, we performed the production of larger membrane areas using the DS method. As shown in Supplementary Fig. 13, instead of one 4-cm long fiber substrate in the original autoclave, we increased the number of substrates to four. Also, we employed a large autoclave, which allowed us to place two 8-cm long fiber substrates. Thus, in both autoclaves, the production area increased fourfold. The experimental results given in Supplementary Fig. 13 show that the membrane morphologies, PWP and MR rejections remain almost unchanged compared with the results shown in Figs. 2 and 5, suggesting that the proposed method holds significant potential for large-scale MOF-303 membrane fabrication. Finally, a comparative analysis of the data in Supplementary Table 3 demonstrates that the nanofiltration performance of the DS membranes is competitive with that of the state-of-the-art MOF membranes reported to date. This finding underscores the importance of structural optimization, particularly the alignment of one-dimensional permeation channels, in maximizing membrane performance.

## Discussion

In this study, we experimentally demonstrated well-crystallized MOF-303 membranes with a controlled crystal orientation on the (011) facet by proposing a DS method. The MOF-303 membrane with a one-dimensional pore structure (011) (i.e., the DS membrane) provided straight water permeation channels with ultrahigh water flux for ethanol dehydration, twofold greater than that of the RS membrane with divergent crystal orientations (1-2-2) and 1–2 order(s) of magnitude greater than those previously reported for other water-stable MOF-based membranes.

The pure water permeance data from dye nanofiltration tests also proved the critical role of tailoring the crystal orientation in the fabrication method. The DS membrane showed much greater water

permeance than the other membranes while retaining a very good MWCO (~269). The excellent durability of the DS MOF-303 membrane under pressurized conditions was realized by the better interfacial interaction between the MOF-303 crystals and the HF substrate facilitated by aluminum metal sites available on the alumina support. We believe that the findings from this study can provide valuable insights into the design and fabrication of stable, high-performance MOF-based membranes for water separation applications.

## Methods

### MOF-303 HF membranes fabrication

**One-step hydrothermal method.** 145 mg of $H_3PDC$, 200 mg of $AlCl_3\cdot6H_2O$, and 67 mg of NaOH were added to a Schott glass bottle with 30 mL of deionized water. The glass bottle was ultrasonicated for 15 min before being transferred to a Teflon-lined stainless-steel autoclave. Before the addition of solution, one HF was sealed at both ends with PTFE tape and positioned vertically in a custom PTFE holder, exposing an effective area of 2.1 cm². The synthesis process was carried out in an oven at 100 °C for 48 h. After cooling to room temperature, the prepared samples were removed, washed with methanol three times and activated in a vacuum oven at 100 °C overnight.

**RS method.** In the seeding step, 145 mg of $H_3PDC$ and 67 mg of NaOH were dissolved in 30 mL of deionized water under ultrasonication for 15 min. The solution was transferred to an autoclave in which 1 piece of PTFE-sealed α-alumina HF was placed vertically using a custom PTFE holder. After the solution was added, the autoclave was promptly placed in a 100 °C oven for 2 h. The seeded alumina HF was washed with methanol three times and dried at 100 °C under vacuum to remove unreacted precursors. The secondary growth procedure of the seeded support was performed similarly to the one-step growth method.

**DS method.** This method was similar to that of RS except that 100 mg of $AlCl_3\cdot6H_2O$ was added to the synthesis solution for the seeding process, and the solution for the secondary growth step was the same as that for the one-step hydrothermal process.

### Structural characterization

The crystalline structure of the MOF-303 powder and the crystal orientation of the MOF-303 membrane were characterized by XRD (X'Pert PRO, Cu Kα, Panalytical) at 2θ values between 5° and 50° at 40 kV and 40 mA. The surface morphology and cross-sections of the MO-303 membranes were observed by scanning electron microscopy (SEM, LEO Gemini 1525 FEGSEM). The surface area, pore size, and pore volume of MOF-303 were determined by $N_2$ sorption isotherms measured at 77 K using a physisorption analyser (Micromeritics, 3 FLEX). The wettability of the membrane surface was assessed by recording the water/ethanol contact behavior by employing the Ramé-hart Model 590 Advanced Automated Goniometer. For each trial, 5 μl of DI water or ethanol was dispensed onto the flat membrane surface.

### Vapor sorption measurement of MOF-303

Water vapor and ethanol sorption properties of MOF powder were measured using a DVS Advantage System[49]. A small amount of the dried powder (8–10 mg) was loaded into the sample pan and placed into a sample chamber. To conduct the analysis, vapors were introduced at P/Psat = 90% at 60 °C using compressed air (200 mL min⁻¹) as a carrier gas. Sorption equilibrium was assumed to be reached at each stage when the rate of mass change (dm dt⁻¹) was less than 0.002%/min for 5 min.

### Performance evaluation of MOF-303 HF membranes

The ethanol and 1-butanol dehydration performances were evaluated by a custom-built pervaporation apparatus. Before the HF membranes were immersed in the feed solution, one end of the membrane was connected to the vacuum pump, while the other end was sealed with

epoxy. The feed mixture was heated and maintained at 70 °C during the measurement. On the downstream side, a trap cooled with liquid nitrogen was used to collect the permeated vapor. The collected permeate was analysed via gas chromatography (GC). The separation factor (α) of component $i$ was calculated using the following equation:

$$\alpha_{i/j} = \frac{y_i(1-x_i)}{x_i(1-y_i)} \qquad (1)$$

where $x_i$ and $y_i$ refer to the weight fractions of component $i$ in the feed and permeate solutions, respectively.

The pure water permeance and dye rejection performance were investigated using a homemade dead-end test filtration device. The MOF-303 HF membrane was mounted into a stainless-steel cylinder filled with 80% deionized water. Under a constant pressure of 5 bar, pure water permeated from the membrane was collected and weighed. The pure water permeance ($J$, LMH bar$^{-1}$) was calculated with the following equation:

$$J = \frac{V}{A \times t \times P} \qquad (2)$$

where $V$ is the permeate volume, $t$ is the test time, $P$ is the test pressure, and $A$ refers to the effective test area of the membrane.

Three neutral dye dispersions, including azobenzene, MR and PPh-IX, at a concentration of 20 ppm in deionized water were utilized as feed to evaluate the nanofiltration performance of the membranes. To avoid the effect of dye adsorption by the membranes on the rejection performance, all the samples were immersed in the feed solution for 24 h before the rejection test. After reaching static equilibrium, the dye rejection test was conducted with the same dead-end test system. Permeants from the membrane were collected after 1 h of the test to avoid the influence of extra adsorption of the membrane under pressure. The dye concentrations in the feed ($C_p$) and permeate ($C_f$) were tested by an ultraviolet–visible (UV-Vis) spectrophotometer (Metller Toledo, UV5), and the rejection ($R$) of dyes was calculated using the following formula:

$$R = \left(1 - \frac{C_p}{C_f}\right) \times 100\% \qquad (3)$$

To determine the concentration of dyes in the permeate, a calibration curve was prepared for each individual dye. A series of dye dispersions with known concentrations (16 ppm, 12 ppm, 8 ppm, 4 ppm, 1 ppm, and 0.5 ppm) were created, and the peak intensities were measured at the corresponding wavelengths. Subsequently, the calibration curves correlating dye concentration with peak intensity were used to calculate the dye concentration in the permeate.

Various materials used to make MOF-303 powders and membranes, along with descriptions of other methods used and various characterization techniques applied to the prepared powders and membranes, are detailed in the SI.

## Data availability
The main data presented and analysed in this project are contained within the article and supplementary information. The data generated in this study are provided with this paper in the source data file. Source data are provided with this paper.

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

## Acknowledgements

The authors gratefully acknowledge the financial support of this project by EPSRC in the United Kingdom (Grant no EP/R029180/1) and the Membrane Material Synthesis for High Selectivity (SynHiSel) project (Grant no EP/V047078/1). We also acknowledge the Barrer Center for the experimental support, and we appreciate the use of equipment for characterization in the analytical laboratory in the chemical engineering department and the electron microscopy center of Imperial College London.

## Author contributions

K.L. conceived the idea. K.L., M.Z., and F.M. designed the research. K.L. led the project and supervision. F.M. and M.Z. constructed the synthesis and test devices. M.Z. prepared the membranes and characterized the materials. M.Z. and F.M. performed the performance tests and data analysis. F.M. and T.G. performed the DVS experiments. M.Z., F.M., J.Y.Y.H., and T.G. discussed the DVS results. M.Z., F.M., and K.L. contributed to the design of the manuscript; M.Z. and F.M. wrote the manuscript; K.L. revised the manuscript; and all the authors reviewed it.

## Competing interests
The authors declare no competing interests.
