## [Transparent Peer Review file · Nature Communications]

Facile orientation control of MOF-303 hollow fibre membranes by a dual-source seeding method

Corresponding Author: Professor Kang Li

Version 0:

Reviewer comments:

Reviewer #1

(Remarks to the Author)

In this manuscript, a novel dual-source seeding (DS) strategy was proposed to fabricate water-stable MOF-303 membrane with a preferred crystallographic orientation on the (011) plane. The DS approach was achieved by the combination of reactive seeding (RS) and conventional hydrothermal methods, wherein the alumina substrate acts as a metal source to react with organic linkers to form seeds, providing better interfacial interactions between MOF crystals and substrate, the second metal source provided by the synthesis solution guarantees the density of seeds. The preferred crystallographic orientation endows MOF membrane with abundant straight one-dimensional permeation channels, resulting in an outstanding water permeation performance. Additionally, the form mechanism of oriented MOF-303 membranes was systematically investigated. The idea to construct oriented MOF membrane by a dual-source seeding method is interesting, which would inspire fabricating oriented polycrystalline structure as next-generation membranes for molecular separation. This work represents a notable step in the field of polycrystalline membrane. I would like to recommend its publication in Nature Communication.

There are some minor comments to be considered as following:

1. In Abstract section, why does MOF-303 material have an excellent water stability. I would suggest the authors add more explanation.
2. It can be seen from Fig. 4a and b, the DS MOF-303 membrane exhibits a total flux of $\sim 18 \text{ kg m}^{-2} \text{ h}^{-2}$ for ethanol dehydration, which is much higher than that for butanol dehydration. Why is there such a big difference since the size difference between ethanol and butanol is only $\sim 0.8 \text{ \AA}$ and butanol is larger than ethanol.
3. The resulting MOF-303 membrane by DS method shows an excellent separation performance for dye rejection with rejection of $\sim 100\%$, the in-depth mechanism is suggested to be provided, size-sieving or adsorption?
4. We can find that a preferred crystallographic orientation (CPO) on the (011) plane provides the straight one-dimensional permeation channels in MOF-303 membrane, leading to a superior water flux, what about the effect of orientation on the membrane pore size?
5. What is the effective membrane area during permeation test?

Reviewer #2

(Remarks to the Author)

The manuscript presents a novel dual-source seeding (DS) method for fabricating metal-organic framework, MOF-303, membranes, with enhanced performance in water purification applications. The authors highlight the limitations of existing MOF membranes on susceptibility to hydrolysis and random crystal orientation in aqueous environments. However, their propose DS method solve the challenge by utilizing a dual-source approach for seeding MOF-303 crystals on an alumina hollow fiber substrate. This approach not only improves the interfacial compatibility between the MOF layer and the substrate but also enables control over the crystallographic orientation, leading to the formation of preferentially tailored membranes.

The performance of the MOF-303 membranes is evaluated in pervaporation and nanofiltration applications. In pervaporation, the membranes exhibit exceptional water flux, surpassing that of most reported zeolite membranes and previous MOF membranes. The authors attribute this superior performance to the straight one-dimensional permeation channels resulting from the controlled crystal orientation. The membranes also demonstrate high water and ethanol selectivity, making them promising for ethanol dehydration processes. In nanofiltration, the MOF-303 membranes exhibit high water permeance and good molecular weight cut-off, effectively rejecting dye molecules. The durability of the membranes is also assessed, and

they show stable performance under pressurized conditions and over extended periods, highlighting their potential for long-term use in water purification systems.

Overall, the manuscript presents a significant advancement in the fabrication of MOF membranes for water purification. The DS method offers a promising solution to overcome the limitations of existing MOF membranes, enabling the development of high-performance, water-stable membranes with tailored crystal orientations. Further research and development in this area could lead to the realization of MOF-based membranes as a viable and efficient technology for addressing the global challenge of water scarcity. This manuscript presents a significant contribution to the field of MOF membranes and suitable to be published in Nature Communications after addressing the suggested comments.

1. The authors attribute the enhanced performance to the controlled crystal orientation, how does the (011) orientation specifically facilitate water transport and selectivity? A more in-depth discussion from structure view of possible mechanism would strengthen the manuscript.

2. A more comprehensive comparison with other state-of-the-art membranes (would provide a better context for the performance of the fabricated membranes).

3. Can the authors try the DS method for large-scale membrane production? This is an important consideration for practical applications.

4. The introduction could benefit from a more concise overview of the challenges and opportunities in MOF membrane fabrication.

5. Some figures could be improved in terms of clarity and labeling. A few typos and grammatical errors should be corrected.

Version 1:

Reviewer comments:

Reviewer #1

(Remarks to the Author)

The authors have well addressed the comments. The revised manuscript is suggested to be published in Nature Communications.

Reviewer #2

(Remarks to the Author)

The authors have carefully addressed the issues. I agreed with the responses from the authors revised and added in the manuscript. It can be published now.

Response Letter to Reviewer Comments

We sincerely thank you for your professional reviews and valuable feedback on our manuscript. According to your constructive comments, which are given in black below, we have made modifications and explanations. In our revised version, changes to the manuscript are highlighted within the documents in red. Point-by-point responses to the concerns from both reviewers are listed below in blue, and quoted texts from the manuscript are in *red and italics*. Moreover, the relevant page and line numbers have been mentioned in this response to guide the reviewers to these changes.

Reviewer #1 (Remarks to the Author):

In this manuscript, a novel dual-source seeding (DS) strategy was proposed to fabricate water-stable MOF-303 membrane with a preferred crystallographic orientation on the (011) plane. The DS approach was achieved by the combination of reactive seeding (RS) and conventional hydrothermal methods, wherein the alumina substrate acts as a metal source to react with organic linkers to form seeds, providing better interfacial interactions between MOF crystals and substrate, the second metal source provided by the synthesis solution guarantees the density of seeds. The preferred crystallographic orientation endows MOF membrane with abundant straight one-dimensional permeation channels, resulting in an outstanding water permeation performance. Additionally, the form mechanism of oriented MOF-303 membranes was systematically investigated. The idea to construct oriented MOF membranes by a dual-source seeding method is interesting, which would inspire fabricating oriented polycrystalline structures as next-generation membranes for molecular separation. This work represents a notable step in the field of polycrystalline membrane. I would like to recommend its publication in Nature Communication. There are some minor comments to be considered as follows:

Thank you very much for your positive and encouraging feedback on our manuscript. Your recommendation for publication in Nature Communications highlights the significance of our work and motivates us to continue advancing this field. We have meticulously addressed the suggested revisions to enhance the clarity and rigor of the manuscript, as outlined below.

Comment 1: In Abstract section, why does MOF-303 material have an excellent water stability. I would suggest the authors add more explanation.

Answer 1: Thank you for highlighting the need for a more detailed explanation of the water stability of MOF-303 in the Abstract. The water stability of MOF-303 can be well explained by Pearson's principle of hard/soft acid/base (HSAB)¹. According to HSAB theory, strong coordination bonds are formed between hard acids and hard bases or

between soft acids and soft bases. Thus, high-valence metal ions (e.g., Fe^{3+} , Al^{3+} , and Zr^{4+}) coordinated with carboxylate-type ligands form robust metal–linker interactions that resist hydrolysis by outcompeting the affinity of metal ions for water molecules. Specifically, in MOF-303, the coordination between high-valent Al^{3+} and carboxylate groups imparts exceptional structural stability. We have added a more detailed explanation of this mechanism to the introduction (Lines 47-58) to strengthen the claims made in the abstract, as shown below:

“According to the hard-soft-acid–base (HSAB) theory¹, stable MOFs are expected to be generated by coordinating high-valence metal ions (Fe^{3+} , Al^{3+} , Zr^{4+} , etc.) with carboxylate-type ligands or soft azolate linkers (imidazolate, pyrazolates, etc.) with divalent metal ions (Zn^{2+} , Cu^{2+} , Ni^{2+} , Co^{2+} , etc.), resulting in strong metal–linker coordination^{2, 3}. By surpassing the affinity of metal ions for water molecules, strong metal–linker coordination bonds are highly resistant to hydrolysis. On the basis of this theory, Zr-based UiO-66 was synthesized as the first MOF to demonstrate good water stability and separation performance in desalination⁴ and dye nanofiltration⁵. Another example is a recently reported Al-based MOF, i.e., MOF-303 [Al(OH)(PZDC), PZDC = 1H-pyrazole-3,5-dicarboxylate], in which the strong metal–linker coordination between high-valent Al^{3+} and carboxylate groups confers high structural stability. Additionally, the pyrazole groups in MOF-303 serve as functional sites, providing high water sorption capacity and facilitating rapid water sorption–desorption cycles, making it a promising candidate for water harvesting applications^{6, 7, 8, 9}.”

Comment 2: It can be seen from Fig. 4a and b, the DS MOF-303 membrane exhibits a total flux of $\sim 18 \text{ kg m}^{-2} \text{ h}^{-2}$ for ethanol dehydration, which is much higher than that for butanol dehydration. Why is there such a big difference since the size difference between ethanol and butanol is only $\sim 0.8 \text{ \AA}$ and butanol is larger than ethanol.

Answer 2: We thank the reviewer for this insightful comment. As mentioned in the manuscript, the permeation rates through MOF-303 membranes are governed by both the adsorption capacity and diffusion rate. The greater flux observed for the ethanol/water mixture than for the water/1-butanol mixture can be explained from both perspectives, which are detailed below.

- 1) Ethanol is more hydrophilic than 1-butanol because of the shorter hydrocarbon part, which provides favourable adsorption of ethanol molecules onto the MOF-303 membrane.
- 2) Due to the -OH and -NH groups in the pyrazole linker, the hydrophilic nature of the MOF-303 membranes preferably traps the first water molecule, which has two hydrophilic ends with three hydrogen bonds¹⁰. The binding of subsequent molecules is influenced by the first water molecule. As a result, the effective aperture size of the MOF-303 membrane during pervaporation is smaller than the theoretical aperture size of approximately 7.7 \AA . Considering the occupancy of the first bonded water molecule (kinetic diameter, 2.9 \AA), the effective pore size is reduced to less than 4.8 \AA . This size, which is larger than ethanol's kinetic diameter (4.3 \AA),

facilitates relatively faster transport through the pores, as depicted in Figure c1. While the kinetic diameter of 1-butanol (5.1 Å) is only 0.8 Å larger, its diffusion is significantly hindered. This hindrance also restricts the transport of water molecules, resulting in reduced flux in the water/butanol mixture. It is well-documented that even a slight variation in molecular size can lead to large differences in diffusion coefficients, which can vary by several orders of magnitude¹¹. A schematic illustration has been added to Supplementary Fig. 7 and mentioned in Lines 258-259 “... which also caused the flux difference between ethanol/water and butanol/water, as illustrated in Supplementary Fig. 7.”

Figure c1 Schematic illustration of the transport of water/ethanol and water/1-butanol mixtures in the MOF-303 membrane.

Comment 3: The resulting MOF-303 membrane by DS method shows an excellent separation performance for dye rejection with rejection of ~100%, the in-depth mechanism is suggested to be provided, size-sieving or adsorption?

Answer 3: We thank the reviewer for the valuable comments. The dye nanofiltration performance test was carried out to evaluate the size-sieving ability of the membrane. Thus, measures have been taken to avoid the influence of adsorption, which can be found in the Methods section, Lines 454-458: “*To avoid the effect of dye adsorption by membranes on rejection performance, all samples were immersed in feed solution for 24 hours before the rejection test. After reaching static equilibrium, the dye rejection test was conducted with the same dead-end test system. Permeants from the membrane were collected after 1 hour of the test to avoid the influence of extra adsorption of the membrane under pressure.*” The nanofiltration performance test results indicated that dye rejection is dependent primarily on molecular size, with larger dyes exhibiting higher rejection rates and smaller dyes showing lower rejection rates. In contrast, the rejection efficiency driven by membrane adsorption is largely influenced by the chemical and physical affinity between the dye and the adsorbent, which is independent

of dye size^{12, 13}.

To further verify the dye removal mechanism, the MR rejection of the DS membrane was tested, and the dye concentrations in the retentate and permeate are presented in Figure c2. The concentration changes in the retentate during the filtration test, after reaching equilibrium, was monitored and is shown below. The increase in MR concentration in the retentate implies that the dye is being separated on the basis of the size-sieving mechanism rather than adsorption by the membrane. This information has been added to Lines 323-325 of the revised manuscript and Supplementary Fig. 8. *“The size-sieving mechanism, rather than adsorption, was confirmed by the increased dye concentration in the retentate, as shown in Supplementary Fig. 8.”*

Figure c2 Concentration changes during long-term MR rejection tests of DS membranes

Comment 4: We can find that a preferred crystallographic orientation (CPO) on the (011) plane provides the straight one-dimensional permeation channels in MOF-303 membrane, leading to a superior water flux, what about the effect of orientation on the membrane pore size?

Answer 4: We appreciate the reviewer’s insightful comments, which highlighted the distinction between membrane permeation channels and MOF pores. As mentioned in the manuscript (Line 59) and illustrated in Figure c3 below, MOF-303 crystals comprise one-dimensional (1-D) pores with a pore size of 0.6–0.8 nm, which determines the separation performance of membranes in the three-dimensional (3-D) structure. In the case of DS membranes with the (011) orientation, 1D pores are perpendicular to the substrate and provide straight permeation channels. In other words, the DS membranes provide a perpendicular one-dimensional channel to the substrate with a pore size similar to that of the MOF-303 pores. In contrast, in the RS membrane with oriented growth on the (1-2-2) planes (i.e., divergent orientation), the MOF-303 crystals overlap or are intersected by neighbouring crystals; therefore, 1D pores may be partially blocked, leading to a smaller pore size and tortuous permeation pathways, ultimately resulting in reduced pore sizes and a lower pore density, as reflected by a lower flux but

slightly higher rejection (or separation factor).

Figure c3 Schematic illustration of membranes formed by different crystal orientations

Comment 5: What is the effective membrane area during permeation test?

Answer 5: We added the dimensions of the alumina HF in the supplementary information, Lines 30-31, as follows: *“The sintered HFs with a diameter of 2.23 mm were cut into 4 cm pieces and then washed with acetone and ethanol to remove impurities.”* The exact effective area has been provided in the revised manuscript which can be found in Lines 395-397, *“Before solution addition, one HF was sealed at both ends with PTFE tape and positioned vertically in a custom PTFE holder, exposing an effective area of 2.1 cm².”*

Reviewer #2 (Remarks to the Author):

The manuscript presents a novel dual-source seeding (DS) method for fabricating metal-organic framework, MOF-303, membranes, with enhanced performance in water purification applications. The authors highlight the limitations of existing MOF membranes on susceptibility to hydrolysis and random crystal orientation in aqueous environments. However, their propose DS method solve the challenge by utilizing a dual-source approach for seeding MOF-303 crystals on an alumina hollow fiber substrate. This approach not only improves the interfacial compatibility between the MOF layer and the substrate but also enables control over the crystallographic orientation, leading to the formation of preferentially tailored membranes. The performance of the MOF-303 membranes is evaluated in pervaporation and nanofiltration applications. In pervaporation, the membranes exhibit exceptional water flux, surpassing that of most reported zeolite membranes and previous MOF membranes. The authors attribute this superior performance to the straight one-

dimensional permeation channels resulting from the controlled crystal orientation. The membranes also demonstrate high water and ethanol selectivity, making them promising for ethanol dehydration processes. In nanofiltration, the MOF-303 membranes exhibit high water permeance and good molecular weight cut-off, effectively rejecting dye molecules. The durability of the membranes is also assessed, and they show stable performance under pressurized conditions and over extended periods, highlighting their potential for long-term use in water purification systems. Overall, the manuscript presents a significant advancement in the fabrication of MOF membranes for water purification. The DS method offers a promising solution to overcome the limitations of existing MOF membranes, enabling the development of high-performance, water-stable membranes with tailored crystal orientations. Further research and development in this area could lead to the realization of MOF-based membranes as a viable and efficient technology for addressing the global challenge of water scarcity. This manuscript presents a significant contribution to the field of MOF membranes and suitable to be published in Nature Communications after addressing the suggested comments.

We are grateful for the reviewer's recognition of the novel aspects of our work. His/her encouraging remarks further motivated us to continue exploring the potential of MOF-based membranes to address global water scarcity challenges. We have carefully considered the reviewer's suggestions and have made the necessary revisions to improve the clarity and impact of the manuscript.

Comment 1: The authors attribute the enhanced performance to the controlled crystal orientation, how does the (011) orientation specifically facilitate water transport and selectivity? A more in-depth discussion from structure view of possible mechanism would strengthen the manuscript.

Answer: Thank you for your insightful question and suggestion. We agree that a more detailed discussion of the structural mechanism would strengthen the manuscript. The (011) orientation of the MOF-303 crystals plays a critical role in facilitating water transport by aligning the one-dimensional pores perpendicular to the substrate. This alignment minimizes the tortuosity of the water transport pathways, allowing for more efficient and direct water permeation through the membrane. Additionally, this preferential orientation helps to reduce the number of grain boundary defects, which are often associated with nonselective transport pathways that can compromise the molecular sieving ability of the membrane. A detailed explanation with a schematic illustration has been added to the manuscript in Lines 333-345 and Fig. 5e, which are copied below:

“As schematically illustrated in Fig. 5e, the (011) orientation of the crystals prepared via the DS method was well-intergrown such that the number of grain boundary defects could be effectively minimized. This is in good accordance with the SEM images shown in Fig. 2e, f, especially compared with the one-step membrane presented in Supplementary Fig. 4, where significant gaps and pinholes tend to form at the grain boundaries in the membranes. These defects, which are larger than intracrystalline

pores, create non-MOF transport pathways that enhance flux but compromise molecular sieving. Furthermore, as an anisotropic MOF with one-dimensional pores, the (011) orientation provides shorter mass transport pathways aligned perpendicular to the substrate, facilitating straight water permeation channels with reduced transport resistance (Fig. 5e)¹⁰. Conversely, while the RS membrane also demonstrated high selectivity due to well-intergrown crystals, the inclined orientation and partially blocked pores from neighbouring crystals created longer, more tortuous permeation channels, resulting in significantly lower water permeance.

Figure 5e Schematic illustration of the DS membrane with vertical one-dimensional channels.”

Comment 2: A more comprehensive comparison with other state-of-the-art membranes (would provide a better context for the performance of the fabricated membranes.

Answer 2: Thank you for your valuable comment. We compared the dehydration performance of ethanol and butanol for the MOF and zeolite membranes. On the basis of your feedback, we have updated Supplementary Tables 1 and 2 to: (a) include recently published MOF and zeolite membranes, and (b) provide comparisons with other state-of-the-art membranes made from different materials. Additionally, we have tabulated the comparison of nanofiltration performance with recently published work in Supplementary Table 3, which is summarized in Lines 365-369 of the revised manuscript: *“Finally, a comparative analysis in Supplementary Table 3 demonstrates that the nanofiltration performance of the DS membranes is competitive among the state-of-the-art MOF membranes reported to date. This finding underscores the importance of structural optimization, particularly the alignment of one-dimensional permeation channels, in maximizing membrane performance.”* We believe these updates enhance the clarity and comprehensiveness of the data. Thank you again for your constructive input.

Comment 3: Can the authors try the DS method for large-scale membrane production? This is an important consideration for practical applications.

Answer 3: Thank you for your insightful suggestion regarding the potential for large-scale membrane production using the DS method. In response, we performed the production of larger membrane areas using the DS method. Instead of one 4-cm long fibre substrate in the original autoclave, we increased the substrate numbers to four. Also, we employed a large autoclave, which allowed us to place two 8-cm long fibre substrates. Thus, in both the autoclaves, the production area is fourfold increased. The experimental results given in Fig c4 show that the membrane morphologies, PWP and

MR rejections remain almost unchanged compared to the results shown in Figs 2 and 5 of the original manuscript.

Due to limited access to large-scale facilities, we were only able to experimentally verify the successful fabrication of four 4 cm HF membranes and two 8 cm HF membranes, achieving a fourfold increase in production area compared to synthesizing a single 4 cm HF membrane in the autoclave. However, based on our findings, we believe that larger-scale production is feasible and worth exploring in future work.

We added the additional information in the revised manuscript (Lines 357-365), *“Furthermore, we performed the production of larger membrane areas using the DS method. As shown in Supplementary Fig. 13, instead of one 4-cm long fibre substrate in the original autoclave, we increased the substrate numbers to four. Also, we employed a large autoclave, which allowed us to place two 8-cm long fibre substrates. Thus in both the autoclaves, the production area was fourfold increased. The experimental results given in Supplementary Fig. 13 show that the membrane morphologies, PWP and MR rejections remain almost unchanged compared to the results shown in Figs 2 and 5, suggesting that the proposed method holds significant potential for large-scale MOF-303 membrane fabrication.”*

Figure c4 Samples fabricated at a larger scale. a HFs were cut into lengths of 4 cm and 8 cm for the fabrication of DS membranes in autoclaves of different sizes. b, c and d Cross-sectional views of the DS membrane observed at random positions in samples fabricated in the larger autoclave. e PWP and MR rejection of five samples fabricated in different autoclaves, Among them, DS_#1-4 are 4 cm samples synthesized in a original autoclave, while DS_#5 and DS_#6 are 8 cm samples synthesized in a larger volume autoclave.

Comment 4: The introduction could benefit from a more concise overview of the challenges and opportunities in MOF membrane fabrication.

Answer 4: We appreciate your professional feedback, and in response, we have made modifications to provide a more concise overview of the challenges and opportunities in MOF membrane fabrication. These revisions aim to improve the clarity and focus of the introduction. The key changes and the reasons are as follows:

1. Line 44 "*Although numerous MOF membranes with promising performances have been developed for gas separation*" was modified to "*Despite progress in gas separation applications*" for conciseness and clarity.
2. Line 45 "*...their potential in aqueous media is still limited because...*" This term was modified to "*...their potential in aqueous media remains limited because...*" for brevity.
3. Line 51 "*Based on this theory, Zr-based UiO-66 has been synthesized as the first MOFs demonstrating...*" was modified to "*Following this strategy, Zr-based UiO-66 was the first MOF demonstrate...*" for simplicity.
4. Line 67 "*Although the abovementioned MOF-303 membranes showed good water stability, they suffer from very low water permeance...*" was modified to "*While these studies demonstrate good water stability, the membranes still suffer from low water permeance*" to improve their clarity and smoothness.
5. Line 70 "*The low permeation of MOF membranes is partly due to the random orientation...*" modified to "*A major challenge contributing to low permeance is the random orientation of MOF crystals...*"
6. Line 77 "*leading to structurally unstable MOF-based membranes*" modified to "*making the membranes structurally unstable*" for brevity

Comment 5: Some figures could be improved in terms of clarity and labeling. A few typos and grammatical errors should be corrected.

Answer: Thank you for your valuable feedback. We have carefully reviewed and improved the clarity and labeling of the figures as per your suggestions. Additionally, we have revised the manuscript to address the typos and grammatical errors, making the text more concise and grammatically correct. The specific changes made to the manuscript are listed below:

Line 23 "*MWCO \approx 269*" modified to "*MWCO \approx 269*"

Line 135 "*encapsulate*" modified to "*encapsulating*"

Line 136 "*form*" modified to "*forming*"

Line 150 "*However, some intercrystalline defects were observed...*" was modified to "*However, intercrystalline defects were observed...*"

Line 157 "*The random crystal orientation of the seeding layer was confirmed by the (011), (022) and (1-2-2) peaks, with a stronger intensity for the (011) peak.*" modified to "*The random crystal orientation of the seeding layer was confirmed by the (011), (022), and (1-2-2) peaks, with the (011) peak showing stronger intensity.*"

Line 188 "*As shown in Fig. 3a, b, the MOF-303 particles were seeded at the grain boundaries and grown to cover the gaps among the grains.*" modified to "*As shown*"

in Fig. 3a, b, MOF-303 particles were seeded at grain boundaries and grew to fill the gaps between the grains."

Line 190 "This phenomenon aligns with the theory of preferred crystal growth on a rough surface over a smooth surface." modified to "This phenomenon aligns with the theory that crystal growth is favoured on rough surfaces over smooth surfaces."

Line 216 "*Rooted from scattered small seeds, a divergent crystal structure emerged in all free directions during the secondary growth step (Fig. 3d) until it finally met and connected to its neighbour to form the RS membrane, as shown in Fig. 3f.*" modified to "*Rooted in scattered small seeds, a divergent crystal structure emerged in all free directions during the secondary growth step (Fig. 3d) until it eventually met and connected with neighbouring crystals to form the RS membrane, as shown in Fig. 3f.*"

Line 223 "*To further illustrate the advantages of our DS approach, the MOF-303 HF membrane was also fabricated by the conventional one-step method, and its SEM and XRD results are given in Supplementary Fig. 4.*" modified to "*To further demonstrate the advantages of our DS approach, the MOF-303 HF membrane was also fabricated using the conventional one-step method, with its SEM and XRD results are presented in Supplementary Fig. 4.*"

Line 248 "*A high water sorption capacity and rapid...*" modified to "*High water sorption capacity and rapid*"

Line 277 "*However, with only one hydrophilic end, ethanol exhibited a lower affinity for MOF-303, as evidenced in Supplementary Video 2, wherein penetration lasting more than twice as long as that of water was observed.*" modified to "*However, with only one hydrophilic end, ethanol exhibited lower affinity for MOF-303, as shown in Supplementary Video 2, where penetration took more than twice as long as water.*"

Line 283 "*network of water*" modified to "*water network*"

Line 286 "*...it is expected that the pores are dominated by water molecules.*" modified to "*...water molecules are expected to dominate the pores.*"

Reference

1. Pearson RG. Hard and Soft Acids and Bases. *Journal of the American Chemical Society* **85**, 3533–3539 (1963).
2. Wang C, Liu X, Keser Demir N, Chen JP, Li K. Applications of water stable metal - organic frameworks. *Chemical Society Reviews* **45**, 5107–5134 (2016).
3. Liu X, Wang X, Kapteijn F. Water and Metal - Organic Frameworks: From Interaction toward Utilization. *Chemical Reviews* **120**, 8303–8377 (2020).
4. Liu X, Demir NK, Wu Z, Li K. Highly Water-Stable Zirconium Metal - Organic Framework UiO-66 Membranes Supported on Alumina Hollow Fibers for Desalination. *Journal of the American Chemical Society* **137**, 6999–7002 (2015).
5. Fang S-Y, *et al.* Construction of highly water-stable metal-organic framework UiO-66 thin-film composite membrane for dyes and antibiotics separation. *Chemical Engineering Journal* **385**, 123400 (2020).
6. Hanikel N, *et al.* Evolution of water structures in metal-organic frameworks for improved atmospheric water harvesting. *Science* **374**, 454–459 (2021).
7. Fathieh F, Kalmutzki MJ, Kapustin EA, Waller PJ, Yang J, Yaghi OM. Practical water production from desert air. *Science Advances* **4**, 3198 (2018).
8. Hanikel N, *et al.* Rapid cycling and exceptional yield in a metal-organic framework water harvester. *ACS Central Science* **5**, 1699–1706 (2019).
9. Wang C, Yan X, Liu X. Polycrystalline Metal - Organic Framework Membranes for Pervaporation. *Industrial & Engineering Chemistry Research* **62**, 10787–10799 (2023).
10. Hanikel N, *et al.* Evolution of water structures in metal-organic frameworks for improved atmospheric water harvesting. **374**, 454–459 (2021).
11. Zhang K, *et al.* Exploring the Framework Hydrophobicity and Flexibility of ZIF-8: From Biofuel Recovery to Hydrocarbon Separations. *The Journal of Physical Chemistry Letters* **4**, 3618–3622 (2013).
12. Allen S, Koumanova B. Decolourisation of water/wastewater using adsorption. *J Univ Chem Technol Metall* **40**, 175–192 (2005).
13. Li H-W, *et al.* Construction of self-healable and recyclable waterborne polyurethane-MOF membrane for adsorption of dye wastewater based on solvent

etching deposition method. *Separation and Purification Technology* **320**, 124145 (2023).